# Opposing brain signatures of sleep in task-based and resting-state conditions

Mohamed Abdelhack ®[1], Peter Zhukovsky[2,3], Milos Milic[1], Shreyas Harita[1,4], Michael Wainberg[1,5,6,7], Shreejoy J. Tripathy[1,2,4,5,6], John D. Griffiths[1,2,4,6], Sean L. Hill ®[1,2,4,5,6,8] & Daniel Felsky ®[1,4,6,9,10] ✉

Sleep and depression have a complex, bidirectional relationship, with sleep-associated alterations in brain dynamics and structure impacting a range of symptoms and cognitive abilities. Previous work describing these relationships has provided an incomplete picture by investigating only one or two types of sleep measures, depression, or neuroimaging modalities in parallel. We analyze the correlations between brainwide neural signatures of sleep, cognition, and depression in task and resting-state data from over 30,000 individuals from the UK Biobank and Human Connectome Project. Neural signatures of insomnia and depression are negatively correlated with those of sleep duration measured by accelerometer in the task condition but positively correlated in the resting-state condition. Our results show that resting-state neural signatures of insomnia and depression resemble that of rested wakefulness. This is further supported by our finding of hypoconnectivity in task but hyperconnectivity in resting-state data in association with insomnia and depression. These observations dispute conventional assumptions about the neurofunctional manifestations of hyper- and hypo-somnia, and may explain inconsistent findings in the literature.

The relationships between sleep, neurocognitive processes, and depression are complex and fraught with paradoxes. Patients with major depressive disorder (MDD) present with both hypersomnia and insomnia[1], while acute sleep deprivation has been shown to act as an effective antidepressant[2]. Making a cohesive model more elusive, sleep-related data collected from self-report are notoriously unreliable[3,4], and depressive symptoms may both result from and lead to cognitive deficits[5,6]. A better understanding of these phenomena rooted in neurological mechanisms may facilitate novel targeted therapies.

Observational studies in humans have shown that sleep deprivation is associated with the degradation of attention, working memory,

reward and dopamine processing, emotion discrimination and expression, and hippocampal memory processing[7]. It has also been associated with aberrant activity observed in the visual cortex[8–10], frontoparietal regions[11], and ventral and dorsal attention networks[12], indicating a possible role of sleep in visual cortical processing via top-down attentional circuits during cognitive task performance. Beyond associations with brain function, sleep quantity and quality are also linked to symptoms of mental illness[13–15], and the biological mechanisms of these links have been explored with neuroimaging. While depression is typically associated with symptoms of insomnia[16], atypical depression is associated with hypersomnia[17]. Insomnia is

[1]Krembil Centre for Neuroinformatics, Centre for Addiction and Mental Health, Toronto, ON, Canada. [2]Campbell Family Mental Health Research Institute, Centre for Addiction and Mental Health, Toronto, ON, Canada. [3]Center for Depression, Anxiety and Stress Research, McLean Hospital, Boston, MA, USA. [4]Department of Physiology, Temerty Faculty of Medicine, University of Toronto, Toronto, ON, Canada. [5]Department of Psychiatry, Temerty Faculty of Medicine, University of Toronto, Toronto, ON, Canada. [6]Institute of Medical Science, University of Toronto, Toronto, ON, Canada. [7]Prosserman Centre for Population Health Research, Lunenfeld-Tanenbaum Research Institute, Sinai Health, Toronto, ON, Canada. [8]Vector Institute for Artificial Intelligence, Toronto, ON, Canada. [9]Department of Biostatistics, Dalla Lana School of Public Health, University of Toronto, Toronto, ON, Canada. [10]Rotman Research Institute, Baycrest Hospital, Toronto, ON, Canada. ✉e-mail: daniel.felsky@camh.ca; dfelsky@gmail.com

hypothesized to be a result of hyperarousal states that cause cognitive fatigue and anxiety that could lead to depressive symptoms[18,19], while the relationship between hypersomnia and depression is still unclear. Some studies suggest that lower quality sleep is associated with negative thoughts through decreased connectivity of the amygdala[20]. Insomnia, daytime dozing, and low sleep quality have also been associated with aberrant functional connectivity at rest, especially the default mode network (DMN)[21], with hyper- and hypo-activation in task-based studies[7,21]. Most neuroimaging analyses, however, have been underpowered and yielded heterogeneous results, leading to inconclusive evidence.

Some initial attempts toward decoding these complex relationships at the population level have been made. Cheng et al.[22] found that the association between self-reported poor sleep and depressive symptoms was partly mediated by patterns of functional connectivity at rest. Similarly, Fan et al.[23] found that resting-state brain connectivity was associated with self-reported insomnia and narcolepsy. However, questionnaire-based sleep assessments do not always align with accelerometry-based sleep measures[24,25], and objective sleep quality measures may better capture aspects of the heterogeneity of sleep-related phenotypes[4]. In addition, previous studies have not probed the relationships between neural signatures of disturbed sleep and depressive symptoms. Another study also investigated the associations of sleep phenotypes in association with obesity, cardiometabolic conditions, brain structure, and cognition but did not account for brain activity[26].

Unfortunately, the majority of our current understanding of sleep has come from acute sleep deprivation experiments and individuals suffering from clinical sleep disorders, which cover only fringe conditions in comparison with the general population[7,16,21,27]. This is especially important given that data collected during acute sleep deprivation does not necessarily have the same impact on brain

dynamics as chronic sleep loss or low sleep quality[7]. When considering the lack of concordance of self-report and objective sleep measures, only a few studies have analyzed brain signatures of both subjective and objective measures of sleep simultaneously[28–30]. When considered alone this may result in misleading neurobiological representations; for example, in primary insomnia, objective sleep measures using polysomnography do not align with the subjective report of participants[24,25]. Together, these limitations of the extant literature have left substantial gaps in our understanding of the role of sleep in mental health, and how sleep affects brain function under different conditions.

To address these gaps, we perform a multi-step analysis of two independent cohorts, namely the UK Biobank and Human Connectome Project (HCP). First, we map the associations of both objective and subjective sleep quality with task-based and resting-state measures of brain function in the UK Biobank[31,32]. Aiming to better understand the relationship between the neural correlates of sleep, depressive symptoms, and cognitive function, we then test the correlations of these neural maps in the UK Biobank and HCP. We find seemingly contradictory relationships between change in neural dynamics in resting-state and task-based conditions. Our observations provide different insights into the counterintuitive relationships between depressive symptoms and sleep in the general population.

## Results

### Self-reported and accelerometer-based measures of sleep are weakly correlated

Figure 1 summarizes the analyses performed in our study. We first quantified the pairwise phenotypic partial correlations between our five behavioral measures: sleep quality measured by an accelerometer (duration of longest sleep bout), self-reported sleeplessness/insomnia frequency, self-reported daytime dozing frequency, cognitive ability

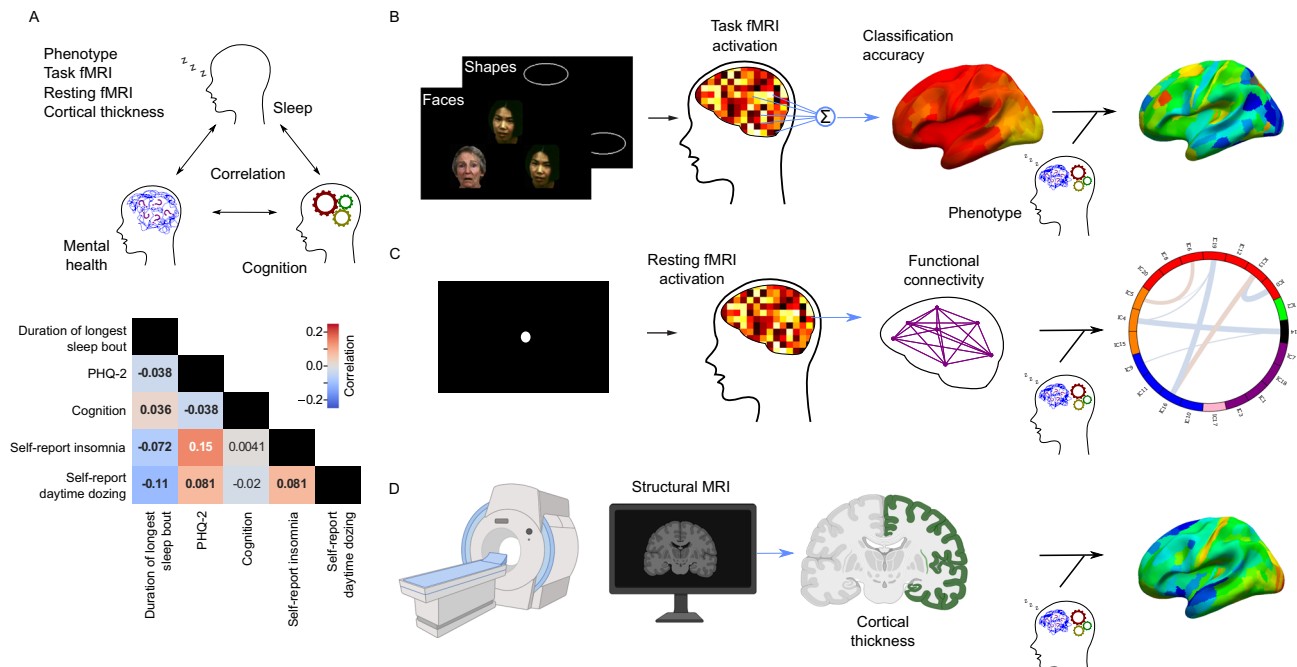

**Fig. 1 | Study Summary. A** shows the partial correlation map between the tested phenotypes of sleep (duration of longest sleep bout, self-reported insomnia, and self-reported daytime dozing), depressive symptoms (PHQ-2 score), and cognition (bolded numbers are correlations significantly different from zero; $p < 0.05/5$). Source data are provided as a Source Data file. **B** shows a summary of the task fMRI experiment, multivariate pattern analysis, and subsequent linear modeling of classification accuracy with selected phenotypes to build cortical maps of

associations (stimulus images obtained with permission from Prof. Deanna Barch). **C** shows a summary of the resting-state fMRI data collection protocol, the calculation of functional connectivity, and the linear modeling to produce a connectivity association map. **D** shows the process for obtaining cortical thickness from structural MRI and linear modeling with phenotypes to generate brain maps similar to those shown in (**B**). Leftmost three figures created with Biorender.com.

measured by a symbol-digit substitution task[33], and subclinical depressive symptoms measured by the PHQ-2[34] with age, sex, study site, ethnicity, socioeconomic status, the difference between the time of accelerometer measurement and assessment center visit, and education as covariates.

Accelerometer-measured sleep quality was weakly correlated with cognitive performance ($r = 0.036$; $p = 5.39 \times 10^{-3}$) while depressive symptoms were correlated with self-reported insomnia ($r = 0.15$; $p = 5.64 \times 10^{-63}$), both in positive directions (Fig. 1A). Self-reported insomnia and daytime dozing frequencies were also positively correlated, though the magnitude of this correlation was similarly very small, with only 0.7% of variance explained ($r = 0.081$; $p = 2.25 \times 10^{-20}$). As expected, the accelerometer-measured duration of longest sleep bout had negative correlations with both self-reported insomnia ($r = -0.072$; $p = 2.21 \times 10^{-15}$) and self-reported daytime dozing ($r = -0.11$; $p = 1.29 \times 10^{-35}$), again with very small effect sizes. Self-reported sleep duration had a moderate but significant correlation with the accelerometer-measured duration of longest sleep bout (Table S5).

## Multimodal neural associations with sleep, depression, and cognition

Having established phenotypic correlations between our measures, we first built a brain map of each phenotype using task-based fMRI (Fig. 1B). We fit multivariate classification models[35,36] using support vector machines (SVM) to classify face and shape trials regardless of task performance. Models from all regions were able to significantly perform above the 50% chance level, however, classifiers using voxels from visual areas were the most accurate (Fig. S2; more details in supplementary results). We carried forward classification accuracies from each region as a proxy for its cortical activation in response to the visual stimuli. We then measured the association of this activation proxy with our phenotypes of interest using ordinary least squares (OLS) regression.

Our measure of task-based brain activation showed significant associations with accelerometer-measured sleep duration, depressive symptoms, and cognitive scores in predominantly visual regions as well as higher multimodal regions in the parietal cortex (Fig. 2A). Cognition also showed significant associations across frontal regions while depressive symptom associations were more global and diffuse (Fig. 2B). Longer sleep bouts were associated with a higher decoding accuracy (stronger multivariate cortical signal), primarily in lateral occipital regions (Highest association region: LO2, $\beta_{normalized} = 0.041$, $p = 6.91 \times 10^{-6}$, $p_{FDR} = 6.13 \times 10^{-4}$; Fig. 2B, S4). These are intermediate processing areas that feed into the ventral stream of vision. Higher regions along the ventral stream showed no significant associations with accelerometer-measured sleep. Depressive symptoms showed significant associations across regions spanning the whole cortex, where higher symptom scores were associated with lower decoding accuracies (Highest association region: PH, $\beta_{normalized} = -0.035$, $p = 1.29 \times 10^{-6}$, $p_{FDR} = 4.63 \times 10^{-5}$; Fig. 2B). The strongest associations were observed in the visual areas, particularly the high-level face-selective and intermediate visual areas (Fig. S4). Higher cognitive scores corresponded to higher decoding accuracy which overlapped with depressive symptoms score effects in visual cortex and prefrontal cortex (Highest cognition association region: PH, $\beta_{normalized} = 0.007$, $p = 3.12 \times 10^{-4}$, $p_{FDR} = 2.35 \times 10^{-3}$; Fig. 2B, C). The latter three phenotypes all had overlapping significant associations in multimodal superior parietal regions (Fig. 2C). These areas are responsible for higher level visual processing of orientation and location as well as motor planning which is reasonable given the nature of the task involving visual recognition and motor action (button pressing). Self-reported insomnia frequency showed no significant effect on the neural coding of visual tasks except in one region in the prefrontal cortex (i6-8, $\beta_{normalized} = -0.003$, $p = 2.21 \times 10^{-5}$, $p_{FDR} = 3.98 \times 10^{-3}$). Self-reported daytime dozing frequency showed no significant associations

in any region. We also tested the associations using univariate analysis of faces vs. shapes contrasts but associations were non-significant for all phenotypes except for cognition (Fig. S5) indicating the distributed nature of signals across regions.

Following task-based analyses, we investigated the associations of resting-state data with our target phenotypes. We first analyzed associations of functional connectivity of independent components across the brain (Fig. 1C), observing many significant associations with accelerometer-measured duration of longest sleep bout that spanned many circuits (Fig. 3A). Daytime dozing showed a similar association pattern with opposite directions of effect due to the inverted scale of the two measures. Insomnia, depressive symptoms, and cognition were associated with only a few circuits and showed little overlap (Figs. S6 and S7; more details in supplementary results). From these associations, we selected one to probe in more detail using seed-based connectivity analysis. Specifically, we investigated the connection between IC5 and IC18 as it showed the strongest association for both duration of longest sleep bout and daytime dozing (duration of longest sleep bout: $\beta = 0.197$, $p = 2.70 \times 10^{-16}$; self-report daytime dozing: $\beta = -0.724$, $p = 4.99 \times 10^{-47}$). The functional connectivity pattern between these two independent components was positively correlated with duration of longest sleep bout and negatively correlated with daytime dozing. We investigated the regions belonging to these two components by selecting the regions that mark higher than the 98th percentile of the component activation. Seed-based results showed a positive association with duration of longest sleep bout and negative association with daytime dozing at the connection level between the posterior side of the inferior frontal junction (IFJp) and almost all the occipital regions in IC18 (Highest association with V3 region, $\beta = 0.006$, $p = 5.24 \times 10^{-5}$). This points to a positive association of sleep bout length with the connectivity between the frontal attentional areas and the intermediate visual regions.

Finally, we investigated the association of each phenotype with cortical thickness (Fig. 1D). Measured duration of longest sleep bout, depressive symptoms, and cognition all showed significant and diffuse associations but with strongest overlap along the auditory, insular, and temporal regions (Fig. S8A; more details in supplementary results). Frequency of daytime dozing showed a sparse pattern that spanned many of the same regions. The results showed that higher cortical thickness was associated with longest continuous sleep (accelerometer-measured), less frequent depressive symptoms, higher cognitive score, and lower frequency of daytime dozing in almost all brain regions This pattern did not hold for the primary visual cortex (V1) and early visual cortex (V2 and V4). Self-reported insomnia did not show any significant association with cortical thickness values.

## Correlations of neural signatures of sleep, depression, and cognition show conflicting relationships under task-activated vs. resting conditions

To quantify the similarity in brain-wide patterns of task-based association between phenotypes, we performed pairwise Pearson correlations between each set of association statistics in the task and resting state conditions. Correlations between neural signatures provide a compact way of visualizing distributed patterns of change in brain dynamics from one phenotype in relation to another. In that context, a positive correlation between two phenotypes means that brain dynamics change similarly in association with each one of them and vice versa. For the task-based condition, in directional agreement with our observed phenotypic correlations (Fig. 1A), the neural signature of accelerometer-measured duration of longest sleep bout was negatively correlated with those for depressive symptoms ($r = -0.62$; $p_{beta} = 5.07 \times 10^{-21}$; $p_{spin} = 1.00 \times 10^{-3}$), frequency of insomnia ($r = -0.18$; $p_{beta} = 0.04$; $p_{spin} = 0.06$), and frequency of daytime dozing ($r = -0.60$; $p_{beta} = 1.00 \times 10^{-21}$; $p_{spin} = 5.00 \times 10^{-4}$). The neural signature for depressive symptoms showed positive correlations with those for

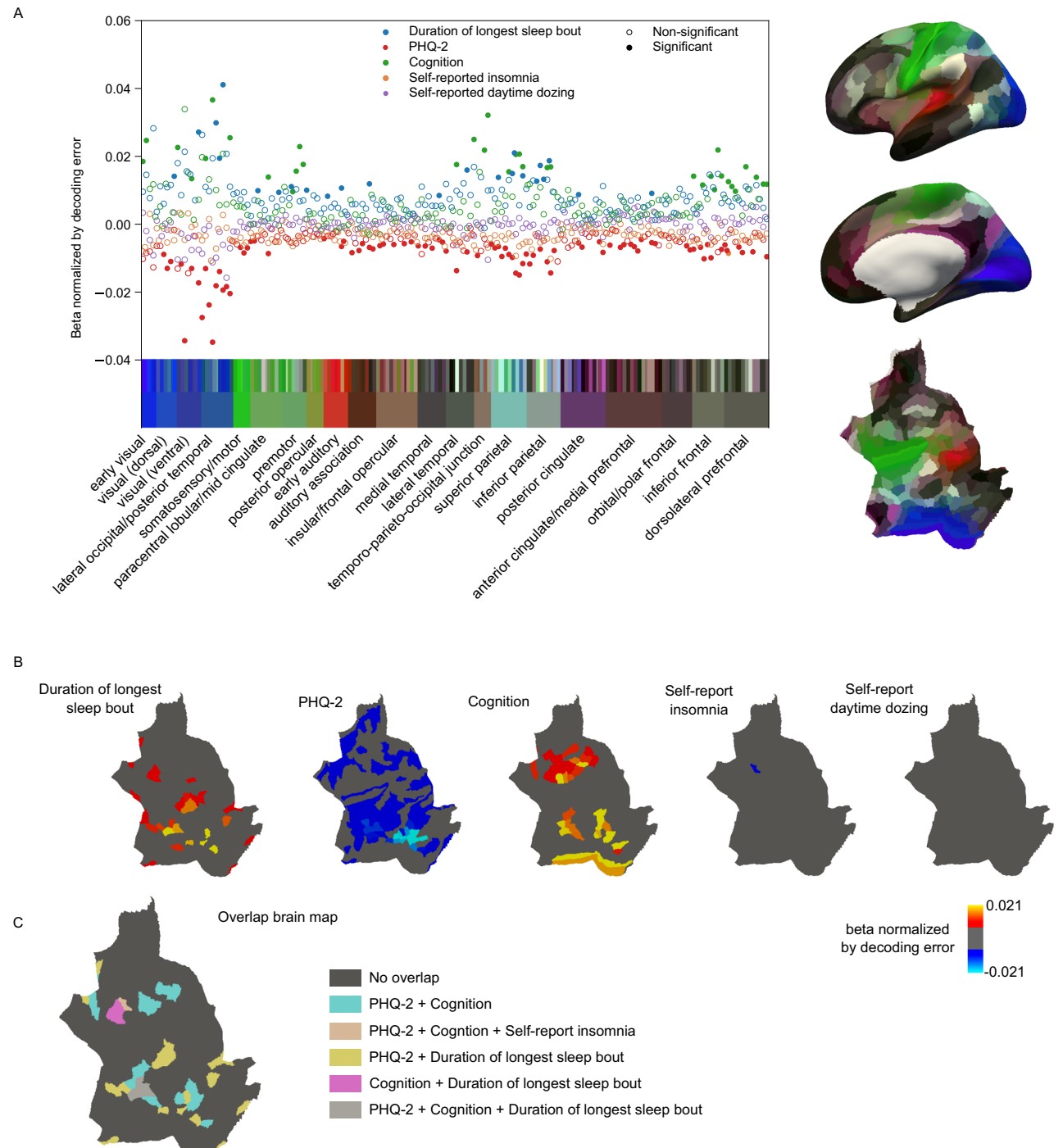

**Fig. 2 | Results of the associations from the task-based fMRI data. A** summarizes the overall beta values of the models normalized by decoding accuracy. The regions are organized and color-coded according to their groupings in the human connectome project[80] where the region color maps are shown on the right. **B** shows the brain region maps with significant associations ($p_{FDR} < 0.05/5$) color-coded by the beta values normalized by decoding accuracy for each of the phenotype models. **C** shows the overlap between regions that showed significant associations with more than one phenotype. Source data are provided as a Source Data file.

both frequency of insomnia ($r = 0.19$; $p_{beta} = 0.03$; $p_{spin} = 0.074$) and daytime dozing ($r = 0.64$; $p_{beta} = 2.36 \times 10^{-22}$; $p_{spin} = 5.00 \times 10^{-4}$), indicating similar effects across the cortex despite the latter two phenotypes showing almost no significant independent associations (Fig. 4A).

Shifting to neural signatures in the resting state condition, a notable difference emerged. In contrast to the results from the task condition and from phenotypic correlations, there were nontrivial positive correlations between the neural signature for duration of

longest sleep bout and those for both self-reported insomnia ($r = 0.59$; $p = 2.71 \times 10^{-21}$) and depressive symptoms ($r = 0.48$; $p = 1.87 \times 10^{-13}$; Fig. 4B). This indicated a similarity between the functional connectivity changes associated with longer continuous sleep, higher frequency of insomnia, and more depressive symptoms - which is counterintuitive. Daytime dozing functional connectivity showed negative correlations with duration of longest sleep bout, self-reported insomnia and depression. To confirm the validity of these results, we retrieved independently modeled associations from Fan et al.[23]

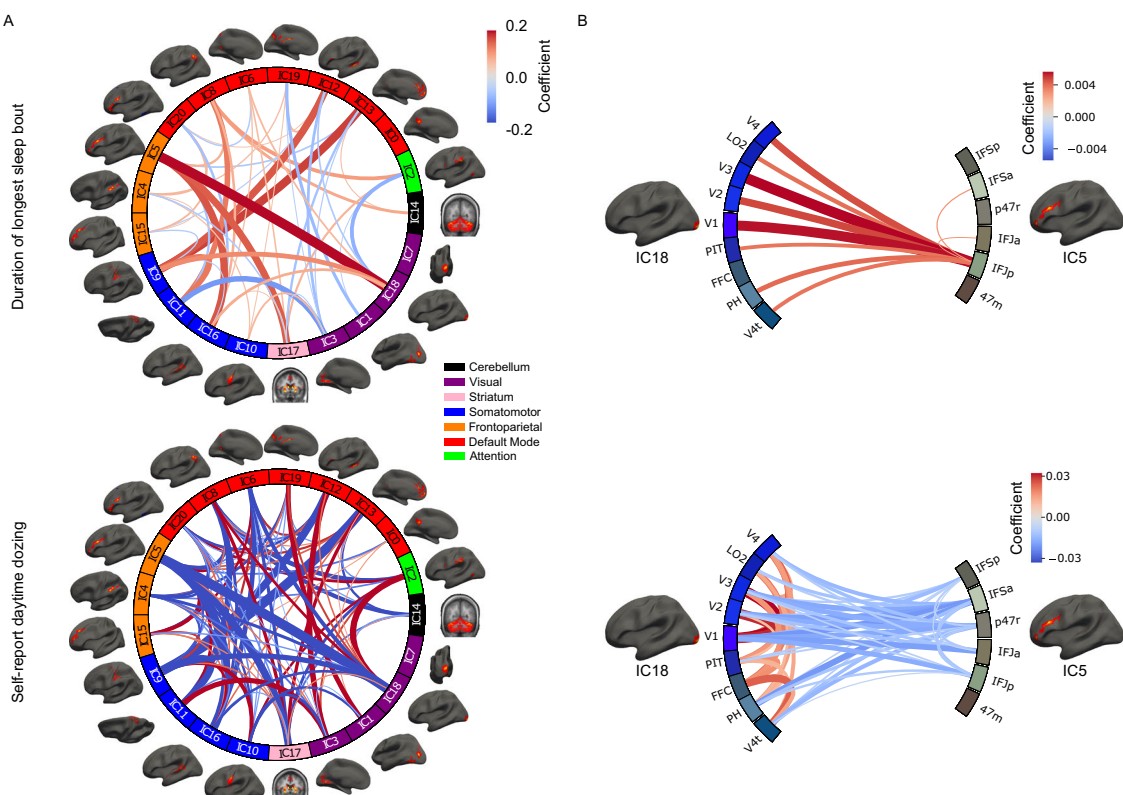

**Fig. 3 | Resting-state connectivity associations results with the sleep phenotypes. A** shows the functional connectivity associations for the accelerometer-measured duration of longest sleep bout and self-reported daytime dozing. The different independent components (IC) are grouped and color-coded based on the Yeo 7 Networks[93]. **B** zooms in on the associations between the different brain regions of IC5 and IC18 showing seed-based correlation associations between different regions belonging to the components of interest. Source data are provided as a Source Data file.

between self-reported insomnia, daytime dozing, and sleep duration (data retrieved at http://www.ig4sleep.org/) and performed the same correlational analyses. Reassuringly, we found nearly identical patterns of correlations between effects (Table S4); self-reported frequency of insomnia and daytime dozing had a correlation coefficient of −0.66 (similar to −0.66 in our analysis). While they did not test accelerometer-measured duration of longest sleep bout, the results from self-reported sleep duration were consistent (correlation with self-reported insomnia = 0.59, with daytime dozing = −0.86). To further confirm these findings, we performed a similar analysis on the independent HCP dataset, which included self-reported sleep measurements using PSQI[37], sadness (proxy for depression) measured using the NIH toolbox[38], and cognition measured by the Mini-Mental State Examination (MMSE)[39]. Results for both task-based and resting-state data were largely in agreement, with the exception of neural signatures for cognition measures (Fig. 4C, D). Correlations between the associations of anatomical models were largely consistent with those from the task fMRI experiment (Fig. S8C). Additionally, we tested the correlation of self-reported sleep duration for task, resting-state, and anatomical measures and it was highly correlated with duration of longest sleep but only for the resting-state data with non-significant correlation for task-based and anatomical data (Table S5).

**Discrepant task-activated and resting fMRI signatures of sleep are partly reconciled by varying sleep duration**

In order to investigate the counterintuitive yet durable positive correlation of insomnia and depression with longer sleep in resting state,

we developed two hypotheses to explain it: (1) a subset of individuals reporting higher levels of depressive symptoms drive the discrepancy due to the fact that both oversleeping and insomnia are possible symptoms of depression, and (2) individuals with insomnia and depressive symptoms possess resting-state neural patterns that resemble those with long sleep resulting in a hyperattentive state, preventing them from sleeping.

To test the first hypothesis, we split the participants by their depressive symptoms into those who have a score of 3 or more as the depressed group and those who have a score less than 3 as the not-depressed group[34]. We fitted the models for each group again and the same correlation patterns between phenotypes persisted in the not-depressed group. In the depressed group, the insomnia and duration of longest sleep bout correlation disappeared (Fig. 5A). This could be a factor resulting from the fact that the depressed group was small, with only 944 participants vs. the remaining 29,918.

We then split the participants into approximately equal groups split by the duration of longest sleep bout median value (greater or less than 6.8 h). Individuals with an average of less than 6.8 measured hours of continuous sleep were labeled "short sleepers", and those with an average of greater than or equal to 6.8 h were labeled as "long sleepers" (Fig. 5C). The positive correlation between sleep duration and both self-reported insomnia and depressive symptoms persisted only within the long sleepers (insomnia: $r = -0.73$; $p = 4.41 \times 10^{-36}$; PHQ-2: $r = -0.59$; $p = 3.34 \times 10^{-21}$). In the short sleeper group, we observed no significant correlation between neural signatures of sleep duration and that for PHQ-2 ($r = -0.079$; $p = 0.26$), and we found a significant negative correlation of sleep duration with self-reported insomnia ($r = -0.20$;

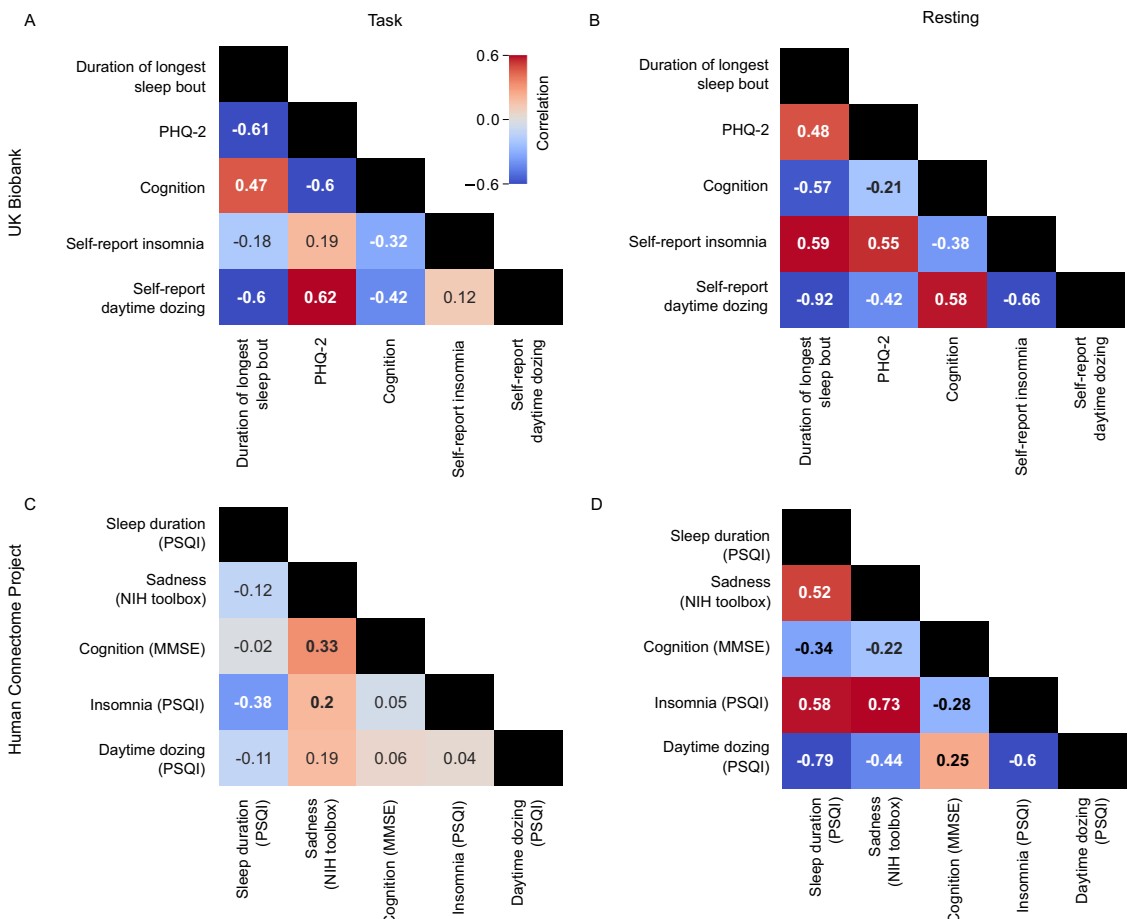

**Fig. 4 | Task and resting conditions show a discrepancy in neural association correlations of sleep, cognition, and depression across two datasets. A** shows the pairwise correlation values between coefficients from each phenotype model of task-based activations across all brain regions in the UK Biobank. **B** shows the pairwise correlation values between coefficients from each phenotype model of resting-state activations across all brain regions of the UK Biobank. **C** shows the task-based pairwise correlations similar to A but for the HCP dataset. **D** shows the resting state pairwise correlations similar to B but for the HCP dataset. Bolded values are statistically significant ($p < 0.05/5$; Bonferroni's correction for five phenotypes). Statistical testing is based on an adaptation of a two-sided student $t$-test for Pearson's correlation values using beta distributions. Source data are provided as a Source Data file.

$p = 0.003$). The positive correlation between signatures of depressive symptoms and self-reported insomnia persisted in both short and long sleepers. This result implies that sleep, when measured in "long sleepers", relates to functional connectivity values that change in a pattern similar to increasing symptoms of depression and frequency of insomnia. However, significant negative correlations with the daytime dozing measure persisted in both groups but the negative correlation between duration of longest sleep bout and daytime dozing did not reach statistical significance.

**Brain regions are hyperconnected under the resting condition with depression and insomnia but hypoconnected during the task condition**

In the previous section, we showed similar resting state patterns between depression and insomnia and duration of longest sleep bout. This was in contrast to the results from the task-based data. In order to investigate the directionality of associations of the neural connectivity patterns giving rise to this discrepancy, we compared global connectivity patterns across resting and task conditions. We calculated the representation connectivity patterns for the task condition and used seed-based connectivity from the previous analysis. We then modeled associations with our five sleep, depression, and cognition phenotypes across all pairs of brain regions as well as an aggregate brain-wide average connectivity measure (Fig. 6A, C). We also modeled the association of the network-specific average connectivity (Fig. 6B) in

order to investigate intra- and inter-network connectivity changes. Results show that, for the task condition, there is a predominantly negative association between representational connectivity and depressive symptoms and self-reported insomnia (Fig. 6A) suggesting hypoconnectivity association with these phenotypes. However, in the resting condition, the associations were mostly positive, suggesting hyperconnectivity. Self-reported daytime dozing showed a strongly negative association suggesting a strong hypoactivation in the resting condition. Duration of longest sleep bout had positive associations for both task and resting conditions, especially in the frontoparietal and attention networks but the average effect was not significant (Fig. 6C). While the spatial connectivity association patterns in task and resting state were not directly correlated, the substantial effect was observed on the global mean level (Fig. S10). These results are also consistent with the correlation results (Fig. 4). Results from the network-wise connectivities show a significant positive association between accelerometer-measured duration of longest sleep bout and the DMN inter-connectivity (Fig. 6B). This significant association was also observed in self-reported insomnia but was accompanied by another significant association between the DMN and frontoparietal network (FPN). Similarly, depressive symptoms showed a significant positive association with the DMN-FPN connectivity but not within the DMN. We confirmed our results through running a connectivity-based predictive modeling (CPM) analysis[40] on each phenotype which yielded consistent results (Fig. S9).

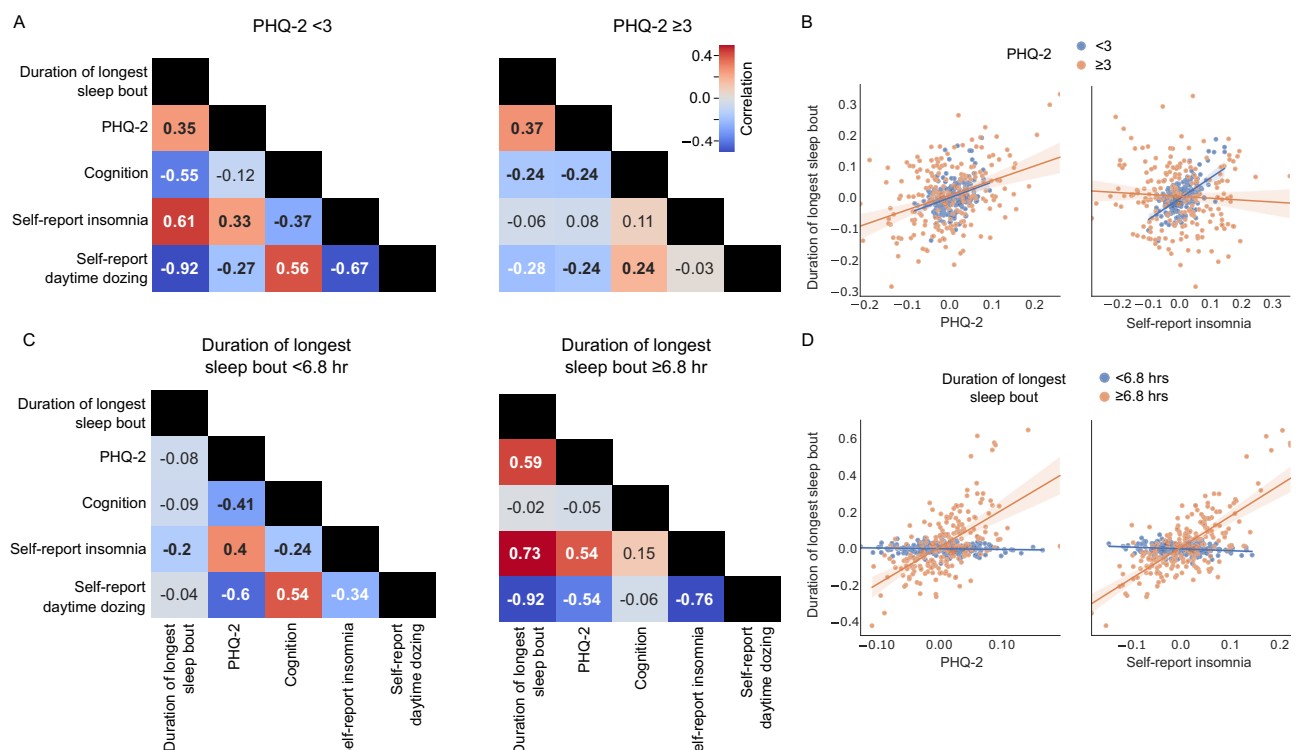

**Fig. 5 | Correlation values of the coefficients of the linear models of functional connectivity values split by PHQ-2 scores and duration of longest sleep bout.** **A** shows the pairwise correlation values for the coefficients of the models split by the PHQ-2 score representing depressed and non-depressed groups. **B** shows a scatter plot with the line of fit between the coefficients of the models for two pairs of phenotypes (duration of longest sleep bout and PHQ-2; duration of longest sleep bout and self-reported insomnia). Error band represent 95% confidence intervals before correcting for multiple comparisons. **C** shows the pairwise correlation values for the coefficients of the models split by the duration of longest sleep bout.

Bolded values are statistically significant ($p < 0.05/5$; Bonferroni's correction for five phenotypes). Statistical testing is based on an adaptation of a two-sided student *t*-test for Pearson's correlation values using beta distributions. **D** shows a scatter plot with the line of fit between the coefficients of the models for two pairs of phenotypes (duration of longest sleep bout and PHQ-2; duration of longest sleep bout and self-reported insomnia). Error band represent 95% confidence intervals before correcting for multiple comparisons. Source data are provided as a Source Data file.

## Discussion

We observed a striking and consistent contrast between the neural representations of objectively-measured and self-reported sleep. Specifically, brain-wide resting state fMRI signatures of long accelerometer-measured sleep were the same as those of higher self-reported frequency of insomnia and depressive symptoms. This seemingly paradoxical result was replicated using summary statistics from a previously published study and in independent analyses of the HCP dataset. Under task conditions, these correlations were inverted. This discrepancy was partially reconciled by showing that the positive correlations in resting state data persisted only for individuals with sleep durations measured on average longer than 6.8 h. Additionally, brain-wide mean connectivity increased with insomnia and depression at resting state but decreased under the task condition. Our findings may explain heterogeneity in existing literature on the neural signatures of sleep and depression, and shed light on the specific circuits responsible for the connections between sleep, depression, and cognition.

Our task-based analyses relied on a measure of signal-based decoding of task trials using machine learning. Superior parietal regions showed significant associations with the duration of longest sleep bout, depressive symptoms, and cognition. Insomnia and dozing showed few significant associations, in line with previous univariate analyses on the same data[23]. Conversely, objective sleep measure revealed associations with neural data sensitivity to neural activity changes in comparison to self-reporting. Duration of longest sleep bout had additional associations with intermediate visual areas at the

lateral occipital junction, with better sleep being associated with higher multivariate activation; these areas are responsible for shape detection[41].

In addition, resting state results revealed widespread associations similar to Fan et al.[23], especially for daytime dozing. We found that functional connectivity between the FPN and in particular the IFJp and lateral occipital regions was positively associated with duration of longest sleep bout. IFJp is known to be responsible for top-down attention[42,43]. This suggests an effect of sleep on the top-down visual attention connections leading to degraded visual processing. It is known that top-down attention can modulate visual cortex activation patterns[44] and thus any impairment in this connection could impair visual function. This effect was reported previously in patients with primary insomnia[45,46]. Previous experiments of sleep deprivation have shown a decreased connectivity between frontal and parietal regions with the visual cortex[47-49] and a decrease in activation of the visual cortex[8-10] that was reversible using trans magnetic stimulation[50,51]. It challenges the results from previous sleep deprivation studies that report a decrease in attention signal at the source at the dorsolateral prefrontal cortex[47-49,52-55] suggesting instead a connectivity impairment. These studies relied mostly on acute sleep deprivation that could lead to transient impairment in cognition as opposed to sustained low sleep quality where connectivity becomes impaired as a result of sustained low attentional signal from the source.

Our central finding was that functional connectivity signatures were positively correlated between longer bouts of

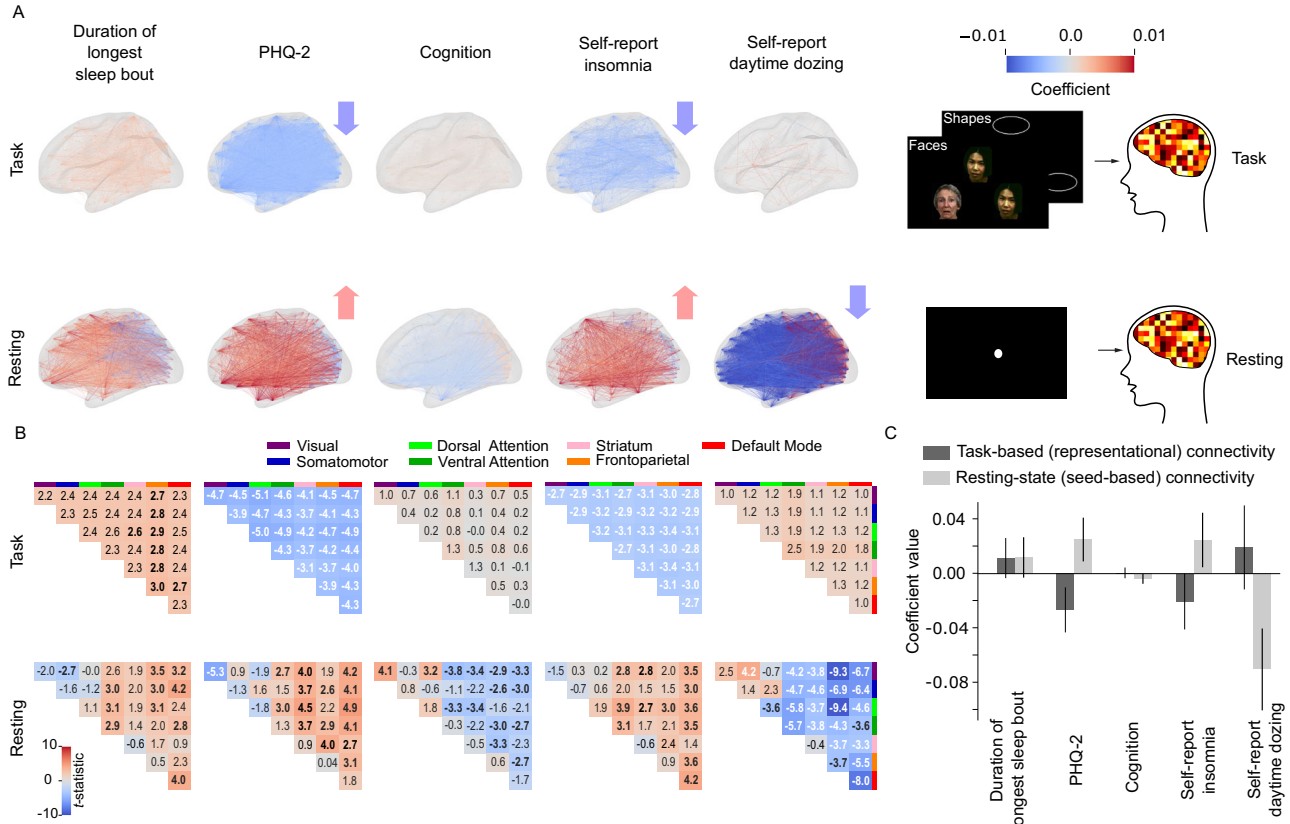

**Fig. 6 | Brain regions are hyperconnected with PHQ-2 and insomnia in resting condition but hypoconnected in task condition. A** shows the significant brain representational and functional connectivity associations with the five phenotypes for each connection between HCP180 regions (*p* < 0.05/5; Bonferroni's correction for five phenotypes). Statistical testing is based on a two-sided student *t* test. **B** Network-wise representational (upper) and functional (lower) connectivity associations (model *t* statistic) with the different phenotypes. Bolded associations are statistically significant. Statistical testing is based on an adaptation of a two-sided student *t* test for Pearson's correlation values using beta distributions. **C** Global mean connectivity (of *n* = 16,110 connectivity values) associations with the five phenotypes. Error bars represent 95% confidence intervals after correcting for multiple comparisons over five measures. Source data are provided as a Source Data file.

accelerometer-measured sleep and both frequency of self-reported insomnia and greater depressive symptoms. This correlation remained in both strata of high and low depressive symptoms, but only persisted in long sleepers when the population was stratified by longest duration of sleep bout. The positive correlation between long sleep and depressive symptoms could in part explain an atypical presentation of depression symptoms: hypersomnia[17]. The positive correlation of long sleep with insomnia could have two explanations: one is that the resting-state signal of a person with a higher frequency of insomnia resembles that of a rested wakefulness state thus preventing them from falling asleep and keeping them in a hyperarousal state[14,16,18,19]. Results from sleep EEG suggest that during sleep, signals resemble a hyperarousal state decreasing the quality of sleep in insomnia[56]. Another possibility is that the objective measure of sleep by accelerometry is not capturing the objective sensation of sleep quality which is reported by primary insomnia patients and polysomnography measurements[24,25]. However, we believe that the first explanation is more likely given the phenomenon of contradictory subjective and objective sleep measure results was observed in polysomnography but not accelerometry measures[28–30,57–59] and that our results were reproduced in the HCP dataset where sleep duration was self-reported[60]. This pattern was also reproducible through analyzing publicly available coefficients from an independent analysis of UK Biobank[23].

In our population, the groups of short sleepers (duration of longest sleep bout <6.8 h) showed an inverted association with insomnia which is reasonable but it signals that insomnia neural signature is multimodal resembling both short and long sleep. There was

no significant association between duration of longest sleep bout and PHQ-2 in that group. The positive associations between duration of longest sleep bout remained consistent between depressed and not-depressed groups while insomnia association was insignificant for the depressed group. The non-depressed group showed identical associations with the whole cohort which could be explained by the fact that the non-depressed group represented the majority of the cohort. These results are consistent with previous studies that show a non-linear relationship between sleep duration and depression with sleep close to seven hours being considered optimal[61–63]. Brain-wide mean connectivity results revealed that insomnia and depression are associated with hypoconnectivity in the task condition and hyperconnectivity during the resting condition. Previous studies have found similar results of hypoactivation in primary insomnia for task-based fMRI[64,65] while resting state connectivity results in the literature were mixed[46,66]. For depression, hyperconnectivity was observed in various networks for resting conditions[67,68]. In addition, the sleep state is associated with a breakdown of cortical effective connectivity[69,70] so insomnia being associated with hyperconnectivity in the resting state could signal a reverse effect. In contrast, duration of longest sleep bout showed a sub-threshold increase in average connectivity in both task and resting conditions. This could signify the usual rested wakefulness state with resting-state connectivity being only slightly higher in selected networks related to attention. At the same time, task connectivity is also slightly higher, signifying a better recruitment of neural resources under cognitive load. This explanation is further supported by the Synaptic Homeostasis Hypothesis (SHY)[71], which suggests that a

net increase in synaptic strength and excitability occurs during wakefulness.

Our study has several limitations. First, we studied a general population sample with only a small minority of participants diagnosed with depression, insomnia, dementia, or narcolepsy. Therefore, our findings may not extend to clinical populations with severe impairments and symptoms. Second, while the results of our analyses in UK Biobank and HCP were largely consistent, task signatures of cognition with those for other phenotypes were not entirely consistent. This may have been due to differences in cognitive measures in these two cohorts. The measure of cognition used in the UK Biobank analysis was a word-symbol matching task where changes in performance could indicate cognitive decline. In HCP analyses, we utilized the available test for cognitive decline, the MMSE, but these two measures might not capture the heterogeneity of brain functions that show dysfunction with cognitive decline. This is especially evident in our UK Biobank analysis where the cognitive measure was derived from a single task. Similarly, the results we obtained from our task-based fMRI study might not necessarily generalize to tasks other than face-shape matching which could limit the conclusions of this analysis.

Our results show that longer uninterrupted sleep is related to the strength of sensory and cognitive processing in vision areas, possibly due to the increased top-down attention recruitment. Additionally, we counterintuitively found similarities in resting state activity among people with insomnia, long sleep, and depressive symptoms which could signal hyperarousal in resting state activity. One possible interpretation of the process involved is that hyperarousal increases the possibility of cognitive fatigue that may end up causing a reduction in task-based activation. Additionally, hyperconnectivity during wakefulness sets the stage for the phenomenon of 'local sleep'[72,73] which refers to episodes where specific neuronal groups enter a sleep-like state during wakefulness. These episodes, often triggered by heightened excitability, result in temporary reductions in both neural activity and connectivity. It could elucidate the diminished connectivity we noted during task performance in insomnia sufferers. Importantly, Nir et al.[73] have demonstrated that these local sleep episodes are associated with lapses in cognitive performance, providing a plausible link to the cognitive impairments observed in individuals with insomnia. This hyperarousal, along with failure to allocate neural resources when exposed to cognitive load, could give rise to depressive symptoms. It also highlights the heterogeneity of sleep quality factors where previous studies showed depression to be associated with poor sleep (measured by the overall PSQI sleep score) with functional connectivity in the dorsolateral prefrontal cortex, cuneus, and orbitofrontal cortex mediating the relationship with depression[22]. We showed here that within the same HCP cohort, different components of the PSQI score have different neural signatures.

In summary, the functional hyperconnectivity in resting states observed among insomnia sufferers can be conceptualized as a manifestation of increased synaptic excitability, as outlined by the SHY, potentially exacerbated by sleep deprivation. The concurrent decrease in effective task-related connectivity might be attributed to episodes of 'local sleep', causing temporary disruptions in neural activity and leading to cognitive lapses. This integrated perspective not only aligns with existing scientific literature but also offers a nuanced understanding of the intricate interplay between sleep duration, neural connectivity, and cognitive function in insomnia.

Our study highlights the importance of investigating the multi-modal signature of phenotypes to understand their diverse manifestations that could give rise to similar symptoms. Our results are supported by a large sample size of over 30,000 participants from the UK Biobank and over 800 participants from the HCP. The sheer size of these datasets also allows for studying more brain-wide associations with reproducible quality and relatively accurate effect sizes[74]. We

uncover a phenomenon of brain-wide similarities between sleep quality, insomnia, and depressive symptoms that could guide advancing clinical practice to investigate more fine-grained details of sleep habits to guide the optimal care plans all while concurrently tracking the cognitive load of patients.

## Methods

This research complies with all ethical regulations relevant to this work. The use of UK Biobank data is governed under the under Approved Research Project #61530. UK Biobank has approval as a Research Tissue Bank from the North West Multi-center Research Ethics Committee. The HCP use is governed by the Restricted Data Use Agreement terms. This work was conducted with approval from the CAMH Research Ethics Board.

### Software

We utilized FreeSurfer 6.0.0 and FSL 6.0.5.1 tools for brain region parcellation and label transformation as well as for cortical thickness measurements and seed-based correlation analysis and higher-level modeling of its results. We used python 3.6.8 for subsequent analyses with Brain Decoder Toolbox 2 v0.19 for brain region data extraction, scikit-learn v0.24.1 for SVM classifier construction, statsmodels v0.10.1 for OLS model creation. For plotting and visualization, we used the libraries seaborn[75] v0.9.0, visbrain v0.4.5, matplotlib 3.3.4, and mne-connectivity v0.2. We also used the following general utility libraries: pandas 0.25.1, numpy 1.19.5, and scipy 1.5.3.

### Dataset

Data was obtained from UK Biobank[31,32] application #61530. We collected data for the functional magnetic resonance imaging for the resting state and task-based paradigms as well as the anatomical data. We also utilized the task data from E-Prime software (software data) to characterize the task-based runs. For sleep data, we obtained both data from the self-report sleep quality measures collected at the same imaging instance and from wrist-based accelerometers, which were worn over a 7-day period and used for extracting quantitative measurements of sleep quality[15]. Other psychiatric (PHQ-2) and cognitive measures (symbol digit substitution task) were collected from the self-reported mental health questionnaires and cognitive test results[33] conducted at the same imaging instance. Table S1 the number of valid subjects extracted for each data modality. Covariates were extracted from the demographics data in UK Biobank (sex, age, socioeconomic status, ethnicity, and education level) and the measurement-specific factors (difference in time between accelerometry measurement and brain image acquisition, head motion, face-shape task performance, and measurement site). To maximize the number of participants and strengthen statistical power in each association analysis, we included all participants with an available measurement for each phenotype independently rather than investigating only the participants with all valid measures (Fig. S1). This led to different numbers of participants for each phenotype measurement (Table S2). Table S3 shows the variable codes extracted from the UK Biobank dataset.

The phenotypes included in the study were validated as proxies for sleep, cognition, and depression. Sleep questionnaire data from the UK-Biobank are equivalent to those in the Pittsburgh Sleep Quality Index questionnaire[37] which is a common sleep quality index[76] that was shown to have good internal reliability and validity[77]. The depression symptom questions represent those used to score depression on the PHQ-2 scale[78] which is used to measure depressed mood and anhedonia and was shown to be an effective first-step screening for depression[79]. The measure for cognition used is the digit-symbol substitution test score which is a standard test score in clinical neuropsychology. It was validated in previous studies to be sensitive to changes in cognitive function impacted by many factors including those that are associated with MDD[33].

## Phenotypic correlation analysis

We measured the pairwise phenotypic partial correlations between five output parameters: sleep quality measured by an accelerometer (duration of longest sleep bout), self-reported sleeplessness/insomnia frequency, self-reported daytime dozing frequency, cognitive ability measured by number of correct matches in a symbol-digit substitution task[33], and subclinical depression score measured by the PHQ-2 scale[34] with age, sex, study site, ethnicity, socioeconomic status, difference between time of accelerometer measurement and assessment center visit (only for the accelerometer output parameter), and education level as covariates. We calculated confidence intervals and significance by the 99% confidence intervals to correct for multiple comparisons (0.05 significance level over five outputs).

## ROI-based analysis

Regions of interest for the multivariate pattern analyses were constructed using the predefined cortical parcellations from the HCP[80]. We combined the bilateral regions of interest resulting in 180 parcellations. The labels from the HCP parcellation were transformed using FreeSurfer software[81] from the fsaverage subject cortical surface to each subject's surface in the dataset. Labels were then transformed into the volume space of the fMRI data for each of the resting state and task-based paradigms.

## Task fMRI analysis

The task fMRI experiment in UK Biobank data comprised a modified version of the face-shape matching task[82,83]. In this task subjects viewed a central cue stimulus accompanied by two stimuli on the right and the left with one of them matching the central cue. Subjects were tasked to press a button identifying which of the two stimuli is the one matching the central cue. The trials contained either human faces or 2D shapes (circle, horizontal ellipse, and vertical ellipse). In order to perform brain-wide association analysis with the task-based fMRI data, we built multivariate classification models[35,36] using SVM to classify the face and shape trials regardless of subject's performance. Multivariate methods have an advantage over the readily available univariate analyses in that they select voxels relevant to the task and aggregate their effect leading to more sensitivity to the classification target in cases of distributed coding[84-87]. Models were created for each region of interest where regions were delineated according to the HCP parcellation[83]. We carried forward the classification accuracies from each region as a proxy for its cortical activation in response to the visual stimuli. We then measured the association of classification accuracy with our phenotypes of interest using OLS regression models. We created OLS models relating the classification accuracy of each region and sleep efficiency. We also added the relevant covariates to the model (sex, age, imaging site, head motion, socioeconomic status, education level, ethnicity, task performance accuracy mean, task response time mean, task response time standard deviation, sex and age interaction, and accelerometry time relative to brain acquisition). To correct for multiple comparisons, we adjusted the p-values for the false discovery rate using the Benjamini/Hochberg method. We then divided the resulting model coefficients by the classification error to up-weight regions with voxels most responsive to the stimuli.

## Multivariate pattern analysis

We utilized the readily preprocessed task-based fMRI data from UK Biobank to create classifiers between faces and shapes for each brain region. Time series from each region was extracted using Brain Decoder Toolbox 2 for Python (https://github.com/KamitaniLab/bdpy). We then applied further preprocessing to the data where the data volumes were shifted by 5 volumes (3.675 s) to compensate for the hemodynamic delay. Data was then filtered to remove the slow signal shift along the run, and then samples were normalized by the mean value to extract the percent signal change. We then averaged the

samples belonging to the same classification category within each block to improve the signal-to-noise ratio. Finally, the data points without stimulus were removed and the samples were then randomized. We ended up with 60 data points for the classifier which were then randomized and divided into training and test datasets in a 6-fold cross-validation scheme to ensure the model is not overfitted. For each fold, we trained a binary SVM classifier with a linear kernel to classify the faces and shapes. The mean classification accuracy (Fig. S3) from each region was then calculated and utilized as a proxy for the strength of encoding of stimuli in this brain region.

## Representational connectivity analysis

We extracted and preprocessed the task fMRI data in a similar fashion as in the MVPA analysis. We then divided the stimuli into seven different categories based on the content of stimuli with three categories representing shapes (circle, horizontal ellipse, and vertical ellipse) and four representing faces (male, female, angry, and fearful faces). The voxel data for each of these conditions were then averaged creating a vector of voxel data for each region. We then computed the representational dissimilarity matrix (RDM)[88] for each region. To calculate representational connectivity, we conducted a second-order similarity analysis between region pairs by calculating the Pearson correlation coefficient between the lower triangles of their RDMs.

## Cortical thickness measurement

Cortical thickness was measured for each brain region using Freesurfer software anatomical statistics measurement tools using the FreeSurfer reconstructed brain anatomy images provided by UK Biobank. We then created OLSs models relating cortical thickness data to sleep efficiency and relevant covariates (sex, age, socioeconomic status, education level, ethnicity, imaging site, sex and age interaction, accelerometry time relative to brain acquisition). To correct for multiple comparison, we adjusted the p-values for false discovery rate using the Benjamini/Hochberg method.

## Functional connectivity analysis

We extracted the readily-processed functional connectivity data based on full correlation from the UK Biobank repository (variable code: 25750) and created OLS models relating functional connectivity between each node (independent component) and sleep efficiency. We also added the relevant covariates to the model (sex, age, imaging site, head motion, socioeconomic status, education level, ethnicity, sex and age interaction, and accelerometry time relative to brain acquisition). To correct for multiple comparisons, we adjusted the p-values for multiple comparisons using Bonferroni's correction for five phenotypes and 21 independent components.

## Seed-based correlation analysis

In order to create more fine-grained connectivity patterns that also map to the same regions as the task-based fMRI, we ran a seed-based correlation analysis on each region using FSL dual regression tool[89]. We then divided the resulting correlation map into the HCP region space computing the mean over each region resulting in a 180 × 180 matrix of connectivity. We then normalized the rows of the matrix by the auto-correlation values (diagonal of the matrix). Results were used to construct a higher-level model with sleep efficiency as the independent variable and the resting state covariates similar to the OLS models previously described in the functional connectivity analysis.

## Brain-wide mean connectivity analysis

We calculated brain-wide mean connectivity by averaging the seed-based connectivity across all node pairs from the HCP regions for the resting state data. For the task-based data we averaged the representational connectivity measures across all the regions. We then built OLS models for each mean connectivity value for each

phenotype and calculated the model coefficients and confidence intervals based on a *p*-value of 0.01 based on Bonferroni correction for five phenotypes.

### Second order correlations testing similarity of neural signatures

We correlated the beta coefficients of the models for duration of longest sleep bout, PHQ-2, cognition, self-reported dozing, and insomnia with resting state connectivity and task-based activation maps and cortical thickness in the HCP space (Glasser 2016) with each other. *p*-values were calculated on the probability that the beta distribution is drawn from a distribution with zero correlation. To correct for multiple comparisons we used Bonferroni correction ($p_{beta} < 0.05/5$). Additionally, to correct for auto-correlations for region-based maps in task and anatomical association correlations, we conducted a spin test[90]. Since our results are bilateral, we conducted 1000 permutations of the results with each of the hemispheres and calculated the *p*-values as the percentage of permutations yielding correlation values with a norm higher than the correlation value. We also corrected for multiple comparisons using Bonferroni correction ($p_{spin} < 0.05/5$). We report the correlation value as significant if it is significant for both statistical tests.

### Connectivity-based predictive modeling

To conduct CPM[40], we first regressed out the covariates from the connectivity measures of both the task-based and seed-based connectivity results. We built OLS models for each connectivity value with the covariates and extracted the residuals. These residuals were then used for the CPM analysis, where the correlation between each connectivity value and the phenotype were calculated and only values with *p*-value < 0.01 were selected. The selected values were then averaged and data were divided into training and test sets using 10-fold cross-validation. Model performance was evaluated using Pearson's correlation between predicted and true phenotype in the test set.

### Human connectome project data analysis

We extracted the HCP data from the young adult project[60,91]. We extracted the data for the emotion task and the resting-state. For the emotion task, there were two runs for each subject with an identical task as that of the UK Biobank. We concatenated the data for these two runs and constructed the SVM models similar to the protocol used for UK Biobank. For the resting-state data, we utilized the already processed functional connectivity based on the full correlation between nodes defined by the group-ICA analysis.

For the phenotypes equivalent to those we analysed in the UK Biobank, we used the sleep parameters based on the PSQI sleep score[37] as there was no objective sleep measures. We utilized the self-reported sleep duration as a proxy for the accelerometer-measured duration of longest sleep bout, the PSQI second component that relies on difficulty of falling asleep as a proxy for insomnia, and the answer to the question on trouble staying awake during daytime activities as a proxy for daytime dozing. For depression measure, we used the reported sadness score from the assessment of self-reported negative affect measure from the NIH toolbox[38]. For cognition, despite the HCP data containing cognitive test, the test score we utilized for the UK Biobank was not done for the HCP cohort. We utilized the MMSE results as generic test for cognition[39]. The complete set of subjects with all imaging and behavioral phenotypes available was 807.

### Data availability

This study used the UK Biobank data that was accessed under Approved Research Project #61530 (PI D. Felsky). Access to data is available through an application process. More information about access to UK Biobank data is available here: https://www.ukbiobank.ac.uk/enable-your-research/apply-for-access. The HCP Young Adult dataset is freely available online: https://www.humanconnectome.org/study/hcp-young-adult/data-releases. Source data for all figures in main text are provided as a Source Data file. Source data are provided with this paper.

### Code availability

Codes to analyze this data are available on Github: https://github.com/FelskyLab/sleep_depression_2023 and deposited to Zenodo[92].

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

## Acknowledgements

The authors would like to thank the members of the Whole Person and Population Modeling group members for their valuable support, discussions, and feedback throughout the different stages of this study. We also thank Qing Chang for his help with data access and Prof. Colin Hawco and Dr. Erin Dickie for their helpful discussions over fMRI data processing methodology. We would also like to thank the participants in the UK Biobank and HCP studies for their contribution to open science. We also thank Prof. Deanna Barch for allowing us to use the stimulus images and the UK Biobank support and community for their help with the understanding of data. This study was supported via the generous contribution of grants for D.F. from The Koerner Family Foundation New Scientist Program, The Krembil Foundation, the Canadian Institutes of Health Research, the Canadian Foundation for Innovation, and the CAMH Discovery Fund. Author M.A. was further supported by the CAMH womenmind postdoctoral fellowship.

## Author contributions

Conceptualization: M.A. and D.F. Method development: M.A. and D.F. Data organization: M.M. Data cleaning and analysis: M.A., S.H. and P.Z. Visualization: M.A. and P.Z. Supervision: D.F., S.L.H., J.D.G, S.J.T. Writing - Original Draft: M.A. and D.F. Writing - review and editing: M.A., D.F., S.L.H., S.H., M.W., P.Z.

## Competing interests

The authors declare no competing interests.
