## [Peer Review File · Nature Communications]

Opposing brain signatures of sleep in task-based and resting-state conditionsReviewer #1 (Remarks to the Author):

The manuscript by Abdelhack and colleagues utilized the large cohort of UK biobank to investigate the association between brain function and sleep quality measured by self-report questionnaire and accelerometer. Cognitive function and depressive symptoms were also utilized to characterize the brain-wide patterns. They found neural signatures of insomnia and depression were negatively correlated with neural signatures of sleep duration in the task condition but positively correlated in the resting-state condition. These findings suggested contradictory patterns of resting-state and task-based activation in response to poor sleep. The authors developed clear hypotheses to explain the results and my concerns were with the introduction and the reporting of the findings.

Major:

- 1) The introduction of the manuscript should be carefully revised to be more logical and clearer. The connections of the idea are unclear therefore it's hard to follow the main idea. Meanwhile, the description of previous work on neuroimaging (fourth and fifth paragraph) is seemingly too long compared with the first three paragraphs.
- 2) Considering the classification accuracies were utilized as a proxy for cortical activation in response to the visual stimuli, careful validation should be performed to demonstrate the effectiveness of this index. Meanwhile, the UK biobank also provided the task fMRI data of face activation, why not directly utilize the measure of task-based fMRI provided by UKB?
- 3) The accelerometer-based sleep duration was mainly utilized to measure the association with brain function. Previous studies showed a lower association between self-reported sleep duration and accelerometer-based sleep duration. Could the results in the manuscript be validated by the self-reported sleep duration?
- 4) Previous study showed a nonlinear association between sleep duration and mental health and also cognitive function, the authors also found similar neural pattern between long sleepers and depressive symptoms only in task-based data which were worth well discussing.
- 5) In Figure 6c, sleep duration showed a consistent positive association with brain connectivity both in rest and task state, while insomnia and depressive symptoms showed distinct association direction with brain connectivity in the task and rest state. Please classify the possible mechanisms and discuss them.

Minor:

- 1) The p value of the main results and the adjustments for the multiple comparisons should be briefly stated in the results section. There were no p values of the associations on page 9 and 12.
- 2) Some sentences in the abstract and results are not so clear. Please revise it.

Reviewer #2 (Remarks to the Author):

Abdelhack et al. analyzed the correlations between task and resting-state brain-wide signatures of sleep, cognition, and depression in over 30,000 individuals. They found that neural signatures of insomnia and depression were negatively correlated with neural signatures of sleep duration in the task condition but positively correlated in the resting-state condition, suggesting opposing brain signatures of sleep in task-based and resting-state conditions. Overall, this study is interesting and the manuscript is well-written. The strengths include a large sample size, elaborate methodologies, and validation in an independent sample. I have some concerns during the review.

1. The authors should verify or provide evidence for the validity of the questionnaires they used to assess sleep, cognition, and depression.
2. The flowchart in Figure 1 is a little confusing. Actually, correlations between each neuroimaging modality and all behavioral phenotypes were tested.
3. The reviewer wonders whether the results will be present when other task-based fMRI data are adopted.
4. The authors should specify the rationale for investigating the correlations of neural signatures (i.e., spatial correlations between association statistics) of sleep, depression, and cognition task-activated vs. resting conditions. What did the positive and negative correlation mean?
5. Did the authors consider spatial auto-correlations?
6. The authors observed that brain regions were hyperconnected under the resting condition with depression and insomnia but hypoconnected during the task condition. They may directly examine

the correlations between these affected functional connectivity to show that they were inversely correlated during task and resting conditions.

7. When exploring the correlations between brain connectivity and behavioral phenotypes, connectome predictive modeling (CPM) is strongly recommended.

8. The conclusions (opposing brain signatures of sleep in task-based and resting-state conditions) appear overstated based on the current findings.

In this letter, original reviewer comments are provided in **bold**, and author responses are indicated in normal font. Additions to the text of the manuscript are provided in **green**.

Reviewer #1 (Remarks to the Author):

The manuscript by Abdelhack and colleagues utilized the large cohort of UK biobank to investigate the association between brain function and sleep quality measured by self-report questionnaire and accelerometer. Cognitive function and depressive symptoms were also utilized to characterize the brain-wide patterns. They found neural signatures of insomnia and depression were negatively correlated with neural signatures of sleep duration in the task condition but positively correlated in the resting-state condition. These findings suggested contradictory patterns of resting-state and task-based activation in response to poor sleep. The authors developed clear hypotheses to explain the results and my concerns were with the introduction and the reporting of the findings.

Major:

The introduction of the manuscript should be carefully revised to be more logical and clearer. The connections of the idea are unclear therefore it's hard to follow the main idea. Meanwhile, the description of previous work on neuroimaging (fourth and fifth paragraph) is seemly too long compared with the first three paragraphs.

We have revised and restructured the introduction section to improve its clarity and logical structure. We believe it is now more balanced between concepts and sets up the results section more directly.

Considering the classification accuracies were utilized as a proxy for cortical activation in response to the visual stimuli, careful validation should be performed to demonstrate the effectiveness of this index. Meanwhile, the UK biobank also provided the task fMRI data of face activation, why not directly utilize the measure of task-based fMRI provided by UKB?

Thank you for this comment - we agree that metric validation is important. The use of Multivariate Pattern Analysis (MVPA) is a standard process that was chosen for its improved sensitivity to distributed coding, where the brain stimulus response is distributed among many voxels within a region, compared to univariate analysis that is sensitive to mean change of global signals¹⁻⁴. The latter (univariate) is the one provided by the UK Biobank team. The reasoning behind MVPA's superiority is that it selects the voxels which are most related to the

task within a region and aggregates their contribution. This means it can integrate information from voxels that are considered by the univariate analysis as sub-threshold. Additionally, it will also discard the effect of voxels that are not contributing to the task classification. One disadvantage to this method is its sensitivity to noise⁵, though this concern is mitigated by our approach of cross-validation where the classification accuracy is reported only on withheld test sets which are averaged between folds. To make our motivation more clear, we have added the following text to our methods section citing relevant publications:

“Multivariate methods have an advantage over the readily available univariate analyses in that they select voxels relevant to the task and aggregate their effect leading to more sensitivity to the classification target in cases of distributed coding¹⁻⁴”

To provide additional validation, as requested, we performed confirmation analyses using the univariate features provided directly by UK Biobank. We have added a new figure to the supplementary information (See below, also Figure S8 in supplementary information). As expected, these analyses yielded somewhat fewer significant associations with our selected phenotypes; only cognition showed significant associations with the region-averaged beta values (Figure S8A). Subsequent correlation analysis results were largely consistent with the MVPA-based results except for four values between duration of longest sleep bout and insomnia and between self-reported daytime dozing and PHQ-2, cognition, and insomnia (See figure below, also in supplemental information Figure S8B). Given that the majority of the imaging feature-to-phenotype associations from the univariate analysis were not significant, the correlation values themselves should not be overinterpreted, as they largely represent the product of noise in voxels that are not related to the input and are removed by MVPA.

We have added the following text to the results section to reflect the results from our new univariate analysis:

“Self-reported daytime dozing frequency showed no significant associations in any region. Full table of statistical results is provided in supplementary file 2. We also tested the associations using univariate analysis of faces vs. shapes contrasts but associations were non-significant for all phenotypes except for cognition (Figure S8) indicating the distributed nature of signals across regions.”

Figure S8: Results of associations of task-based fMRI data based on the univariate analysis with the different phenotypes. A summarizes the overall beta values of the models. The regions are organized and color coded according to their groupings in the human connectome project⁶. B shows the pairwise correlation values between coefficients from each phenotype model across all regions.

Out of an abundance of caution, in our main analysis using MVPA, we generally avoid interpreting phenotype relationships for those that did not produce significant associations in the first step. This is why we did not discuss the correlation values between self-reported daytime dozing and insomnia in the task condition despite it also showing inverted relationships with the resting-state; since self-reported daytime dozing did not show significant associations with the task-based decoding accuracy.

The accelerometer-based sleep duration was mainly utilized to measure the association with brain function. Previous studies showed a lower association between self-reported sleep duration and accelerometer-based sleep duration. Could the results in the manuscript be validated by the self-reported sleep duration?

We agree that this is an interesting consideration. Accordingly, we performed additional analyses with self-reported sleep duration and added a new supplementary table of its correlation values with all the five phenotypes in the main text (Table S5; see below). The analysis revealed consistent results for the resting-state data but not the task data, where the results were not significant. We believe this is likely a result of the inaccuracy with which participants are able to recall their own lengths of sleep and is reflected in the low correlation with sleep duration measured by accelerometer. It could also be that continuous sleep is more

important than total sleep duration when determining activation in the task condition. This was also observed in our independent replication dataset from the Human Connectome Project, where sleep duration data were also derived from self-report (i.e. PSQI duration of sleep).

Table S5: Correlation table for self-reported sleep duration with the five main phenotypes for all tested modalities

Phenotype	Correlation of self-reported sleep duration phenotype			
	Phenotypical	Task-based	Resting-state	Cortical thickness
Duration of longest sleep bout	0.116	0.148	0.932	-0.022
PHQ-2	-0.066	0.019	0.455	0.266
Cognition	-0.018	-0.100	-0.621	-0.194
Self-report insomnia	-0.255	-0.068	0.606	-0.141
Self-report daytime dozing	-0.034	-0.090	-0.951	-0.033

Given the above results we have added the following text to the first subsection of results:

“Accelerometer-measured sleep quality was weakly correlated with cognitive performance ($r=0.036$; $p=5.39\times 10^{-3}$) while depressive symptoms were correlated with self-reported insomnia ($r=0.15$; $p=5.64\times 10^{-63}$), both in positive directions (Figure 1A). Self-reported insomnia and daytime dozing frequencies were also positively correlated, though the magnitude of this correlation was similarly very small, with only 0.7% of variance explained ($r=0.081$; $p=2.25\times 10^{-20}$). As expected, the accelerometer-measured duration of longest sleep bout had negative correlations with both self-reported insomnia ($r=-0.072$; $p=2.21\times 10^{-15}$) and self-reported daytime dozing ($r=-0.11$; $p=1.29\times 10^{-35}$), again with very small effect sizes. **Self-reported sleep duration had a moderate but significant correlation with the accelerometer-measured duration of longest sleep bout (Table S5).**”

We have also added the following text to the third subsection of results:

“To further confirm these findings, we performed a similar analysis on the independent HCP dataset, which included self-reported sleep measurements using PSQI²⁶, sadness (proxy for depression) measured using the NIH toolbox³⁷, and cognition measured by the Mini-Mental State Examination (MMSE)³⁸. Results for both task-based and resting-state data were largely in agreement, with the exception of neural signatures for cognition measures (Figure 4C, D).

Correlations between the associations of anatomical models were largely consistent with those from the task fMRI experiment (Figure S7C). Additionally, we tested the correlation of self-reported sleep duration for task, resting-state, and anatomical measures and it was highly correlated with duration of longest sleep but only for the resting-state data with non-significant correlation for task-based and anatomical data (Table S5).”

Previous study showed a nonlinear association between sleep duration and mental health and also cognitive function, the authors also found similar neural pattern between long sleepers and depressive symptoms only in task-based data which were worth well discussing.

We appreciate this comment. Indeed the non-linear effects we identified can be interpreted in the context of previous work. Accordingly, we have added acknowledgement of these effects and relevant citations to the discussion:

“In our population, the groups of short sleepers (duration of longest sleep bout < 6.8 hours) showed an inverted association with insomnia which is reasonable but it signals that insomnia neural signature is multimodal resembling both short and long sleep. There was no significant association between duration of longest sleep bout and PHQ-2 in that group. The positive associations between duration of longest sleep bout remained consistent between depressed and not-depressed groups while insomnia association was insignificant for the depressed group. The non-depressed group showed identical associations with the whole cohort which could be explained by the fact that the non-depressed group represented the majority of the cohort. These results are consistent with previous studies that show a non-linear relationship between sleep duration and depression with sleep close to seven hours being considered optimal⁷⁻⁹.”

In Figure 6c, sleep duration showed a consistent positive association with brain connectivity both in rest and task state, while insomnia and depressive symptoms showed distinct association direction with brain connectivity in the task and rest state. Please classify the possible mechanisms and discuss them.

Thank you very much for this comment. Indeed, we did not comment on this due to the sub-threshold p -values of the average connectivity. However, it is an interesting result that contributes helpfully to the narrative of our study. We have now added a section in the results about possible mechanisms that could be driving these results as follows:

“Self-reported daytime dozing showed a strongly negative association suggesting a strong hypoactivation in the resting condition. Duration of longest sleep bout had positive associations for both task and resting conditions, especially in the frontoparietal and attention networks but the average effect was not significant (Figure 6C). The connectivity association patterns in task and resting state were not related to each other but the effect was observed on the global mean level only (Figure S10).”

Minor:

The p value of the main results and the adjustments for the multiple comparisons should be briefly stated in the results section. There were no p values of the associations on page 9 and 12.

One challenge with presenting these p-values in the results section is that there are many regions explicitly mentioned, so we feel that exhaustively listing statistics would affect readability. To help clarify and improve transparency, we have now added p-values for the top associated regions to the main text. Additionally, for full disclosure, we have added two supplementary files that include all summary statistics from the analyses in Figures 2, 3, and 6. These are referenced as supplementary files 2–6 in the results section.

Some sentences in the abstract and results are not so clear. Please revise it.

Thank you very much for flagging these issues. We scanned the abstract and results sections for unclear phrases and attempted to clarify any ambiguity. We believe the revised text is much improved.

Reviewer #2 (Remarks to the Author):

Abdelhack et al. analyzed the correlations between task and resting-state brain-wide signatures of sleep, cognition, and depression in over 30,000 individuals. They found that neural signatures of insomnia and depression were negatively correlated with neural signatures of sleep duration in the task condition but positively correlated in the resting-state condition, suggesting opposing brain signatures of sleep in task-based and resting-state conditions. Overall, this study is interesting and the manuscript is well-written. The strengths include a large sample size, elaborate methodologies, and validation in an independent sample. I have some concerns during the review.

The authors should verify or provide evidence for the validity of the questionnaires they used to assess sleep, cognition, and depression.

Thank you for this comment that gives us the opportunity to provide valuable context. The questionnaires that were used in the UK Biobank are standards (or equivalent to standards) in their respective fields. Self-reported sleep questions in UK Biobank ask participants to provide general tendencies of insomnia, sleep duration, and daytime dozing. It contains instructions to provide data from the past four weeks in case there is a large variability in those patterns. This is similar to the Pittsburgh Sleep Quality Index¹⁰ where the participant is asked to provide insomnia and daytime dozing habits among other sleep factors for the period in the month before the questionnaire. It is the most common sleep quality index¹¹ and was confirmed by multiple studies to have internal reliability and validity¹². The depression questions represent the

standard PHQ-2 questionnaire that is used to measure depressed mood and anhedonia as a first-step screening for depression^{13,14}. The measure for cognition used is the digit-symbol substitution test score which is a standard test score in clinical neuropsychology. It was validated in previous studies to be sensitive to changes in cognitive function impacted by many factors including those that are associated with major depressive disorder¹⁵.

We have now added further explanation to the methods section (dataset subsection) explaining the validity of each question that we used as shown below:

“The phenotypes included in the study were validated as proxies for sleep, cognition, and depression. Sleep questionnaire data from the UK-Biobank are equivalent to those in the Pittsburgh Sleep Quality Index questionnaire¹⁰ which is a common sleep quality index¹¹ that was shown to have good internal reliability and validity¹². The depression symptom questions represent those used to score depression on the PHQ-2 scale¹³ which is used to measure depressed mood and anhedonia and was shown to be an effective first-step screening for depression¹⁴. The measure for cognition used is the digit-symbol substitution test score which is a standard test score in clinical neuropsychology. It was validated in previous studies to be sensitive to changes in cognitive function impacted by many factors including those that are associated with major depressive disorder¹⁵.”

The flowchart in Figure 1 is a little confusing. Actually, correlations between each neuroimaging modality and all behavioral phenotypes were tested.

Thank you for this comment. We have edited Figure 1 to indicate that correlations are from all modalities and also edited the part with the association icons to reflect all the phenotypes. The updated figure is shown below:

The reviewer wonder whether the results will be present when other task-based fMRI data are adopted.

We agree that one of the acknowledged limitations of our task-based analysis is that it could be task-specific. It can be argued that the task being used here is a very simple one that does not recruit much cognitive ability, so we might expect an even stronger effect from more challenging tasks. This is unless the effect is specific to the cognitive circuits that are employed in this particular task. This limitation is driven by the lack of availability of other task-fMRI datasets in the UK Biobank battery of experiments. Assessing generalizability over different types of tasks is a very interesting question that we plan to address in future work.

The authors should specify the rationale for investigating the correlations of neural signatures (i.e., spatial correlations between association statistics) of sleep, depression, and cognition task-activated vs. resting conditions. What did the positive and negative correlation mean?

Thank you very much for your comment. Investigating the correlation of neural signatures provides a compact way of visualizing distributed patterns of change in brain dynamics from one phenotype in relation to another. In that context, a positive correlation means that brain dynamics change similarly in association with each phenotype.

We added explanations in the results section to justify the rationale of these measures as follows:

“To quantify the similarity in brain-wide patterns of task-based association between phenotypes, we performed pairwise Pearson correlations between each set of association statistics in the task and resting state conditions. **Correlations between neural signatures provide a compact way of visualizing distributed patterns of change in brain dynamics from one phenotype in relation to another. In that context, a positive correlation between two phenotypes means that brain dynamics change similarly in association with each one of them and vice versa.**”

Did the authors consider spatial auto-correlations?

We did consider spatial auto-correlations from multiple perspectives. From a connectivity perspective, we considered the inter-region correlation, and for that, we did normalize our correlation matrices in seed-based correlation analyses by the auto-correlation within each region. We used the term self-correlations to describe this in the methods section. We have now changed the term to “**auto-correlation**” to avoid confusion.

Representational connectivity already has this consideration built into the method since it relies on second order pattern correlation. In the first step, a dissimilarity matrix is constructed for each region by measuring the correlation of average region-wide voxel pattern for each stimulus type (shape-circle, shape-vertical oval, shape-horizontal oval, face-male, face-female, face-angry, and face-fearful). The patterns are averaged across time for each stimulus type. The second step involves calculating correlation between dissimilarity matrices from each region.

Given that the correlation is calculated in a region-wide manner, the auto-correlation is always one.

We also considered the spatial correspondence for associations in different regions for the association maps in the task condition and the anatomical cortical thickness. To confirm that the correlations are not due to a baseline correlation between regions, we conducted a spin test with 1000 permutations for each hemisphere¹⁶. Given our regions are bilateral, we conducted spin tests for each hemisphere independently and then pooled the results. We have now updated our results to include both the p -values for both the correlation (beta) distribution test and the spin test. The results from both tests were highly consistent with only one correlation yielding a significant p -value on one test but not the other. We have also a) added a new subsection in the methods to describe the second order correlations with all statistical testing details as shown below, b) added the spin test p -values to the results section, and c) updated figure 4 (see below) to reflect the values that were significant in both tests.

“Second order correlations testing similarity of neural signatures

We correlated the beta coefficients of the models for duration of longest sleep bout, PHQ-2, cognition, self-reported dozing, and insomnia with resting state connectivity and task-based activation maps and cortical thickness in the HCP space⁶ with each other. p -values were calculated on the probability that the beta distribution is drawn from a distribution with zero correlation. To correct for multiple comparisons we used Bonferroni correction ($p_{\text{beta}} < 0.05/5$). Additionally, to correct for auto-correlations for region-based maps in task and anatomical association correlations, we conducted a spin test¹⁶. Since our results are bilateral, we conducted 1000 permutations of the results with each of the hemispheres and calculated the p -values as the percentage of permutations yielding correlation values with a norm higher than the correlation value. We also corrected for multiple comparisons using Bonferroni correction ($p_{\text{spin}} < 0.05/5$). We report the correlation value as significant if it is significant for both statistical tests.”

Figure 4: Task and resting conditions show a discrepancy in neural association correlations of sleep, cognition, and depression across two datasets. **A)** shows the pairwise correlation values between coefficients from each phenotype model of task-based activations across all brain regions in the UK Biobank. **B)** shows the pairwise correlation values between coefficients from each phenotype model of resting-state activations across all brain regions of the UK Biobank. **C)** shows the task-based pairwise correlations similar to **A)** but for the HCP dataset. **D)** shows the resting state pairwise correlations similar to **B)** but for the HCP dataset. Bolded values are statistically significant.

The authors observed that brain regions were hyperconnected under the resting condition with depression and insomnia but hypoconnected during the task condition. They may directly examine the correlations between these affected functional connectivity to show that they were inversely correlated during task and resting conditions.

Thank you for this comment. Per the reviewer's suggestion, we have investigated the correlations between task and resting conditions. Our results did not reveal opposite correlations between task and resting conditions for insomnia and depression. The reason for

this is that the hyperconnectivity and hypoconnectivity seen in resting and task conditions, respectively, were revealed as an average association, but the spatial pattern of associations was not consistent between the task and resting conditions. We have added a new figure to the supplement (Figure S10, shown below) to illustrate this point graphically. The figure shows a scatter plot of variance-normalized associations between pairs of representational and seed-based connectivity in addition to a regression fit line. The scatter of the associations of insomnia and PHQ-2 can be seen to be shifted to the negative of the axis for the task condition but in the positive direction for the resting condition. The patterns of associations, however, were not consistent, leading to very low correlation values. This could be a result of the difference in methods by which the connectivity is measured between the task and the resting-state conditions, as explained in the response to the last point.

Figure S10: Scatter plot of normalized association values' maps of each connectivity value pair of seed-based and representational connectivity with the colormap representing density of points. Red lines represents the regression line between the resting state and corresponding task-based values.

Based on these results, we have added the following text to our results:

“Self-reported daytime dozing showed a strongly negative association suggesting a strong hypoactivation in the resting condition. Duration of longest sleep bout had positive associations for both task and resting conditions, especially in the frontoparietal and attention networks but the average effect was not significant (Figure 6C). *While the spatial connectivity association patterns in task and resting state were not directly correlated, the substantial effect was observed on the global mean level (Figure S10).*”

Additionally, we corrected an error in figure 6A where the connectivity patterns for PHQ-2 and self-reported insomnia were flipped. That error does not affect any of the conclusions since the associations have similar patterns for those two phenotypes.

When exploring the correlations between brain connectivity and behavioral phenotypes, connectome predictive modeling (CPM) is strongly recommended.

Thank you for this recommendation. We have now included a CPM analysis to confirm our findings, and indeed it showed consistent results. We quantified the number of positive and

negative edges in each model across folds and found the counts of positive vs. negative edges to be consistent with our results for all different variables. We have added these results to supplementary materials (figure is below and in supplementary information Figure S9).

Figure S9: Results of connectivity-based modeling for both seed-based correlation and representational connectivity. A shows the performance of the models based on the correlation between the predicted and true phenotype values in the test set. B shows the number of significant positive and negative edges for each model. Error bars in all figures represent 95% confidence intervals over model permutations.

Based on these results, we have added the following text to the results section:

“We confirmed our results through running a connectivity-based predictive modeling analysis on each phenotype which yielded consistent results (Figure S9). Complete statistical results for both representational connectivity and seed-based correlation are in supplementary files 5 and 6.”

The conclusions (opposing brain signatures of sleep in task-based and resting-state conditions) appear overstated based on the current findings.

Thank you very much for flagging this. We have revised the conclusions based on your suggestions to make the conclusion statements consistent with the findings.

References

1. Pakravan, M., Abbaszadeh, M. & Ghazizadeh, A. Coordinated multivoxel coding beyond univariate effects is not likely to be observable in fMRI data. *NeuroImage* **247**, 118825

- (2022).
2. Davis, T. & Poldrack, R. A. Measuring neural representations with fMRI: practices and pitfalls. *Ann. N. Y. Acad. Sci.* **1296**, 108–134 (2013).
 3. Davis, T. *et al.* What Do Differences Between Multi-voxel and Univariate Analysis Mean? How Subject-, Voxel-, and Trial-level Variance Impact fMRI Analysis. *NeuroImage* **97**, 271–283 (2014).
 4. Jimura, K. & Poldrack, R. A. Analyses of regional-average activation and multivoxel pattern information tell complementary stories. *Neuropsychologia* **50**, 544–552 (2012).
 5. Weaverdyck, M. E., Lieberman, M. D. & Parkinson, C. Tools of the Trade Multivoxel pattern analysis in fMRI: a practical introduction for social and affective neuroscientists. *Soc. Cogn. Affect. Neurosci.* **15**, 487–509 (2020).
 6. Glasser, M. F. *et al.* A multi-modal parcellation of human cerebral cortex. *Nature* **536**, 171–178 (2016).
 7. Bae, S. *et al.* Nonlinear Associations between Physical Function, Physical Activity, Sleep, and Depressive Symptoms in Older Adults. *J. Clin. Med.* **12**, 6009 (2023).
 8. Yin, J. *et al.* Nonlinear relationship between sleep midpoint and depression symptoms: a cross-sectional study of US adults. *BMC Psychiatry* **23**, 671 (2023).
 9. Zhai, L., Zhang, H. & Zhang, D. Sleep Duration and Depression Among Adults: A Meta-Analysis of Prospective Studies. *Depress. Anxiety* **32**, 664–670 (2015).
 10. Buysse, D. J., Reynolds, C. F., Monk, T. H., Berman, S. R. & Kupfer, D. J. The Pittsburgh sleep quality index: A new instrument for psychiatric practice and research. *Psychiatry Res.* **28**, 193–213 (1989).
 11. Mollayeva, T. *et al.* The Pittsburgh sleep quality index as a screening tool for sleep dysfunction in clinical and non-clinical samples: A systematic review and meta-analysis. *Sleep Med. Rev.* **25**, 52–73 (2016).
 12. Fabbri, M. *et al.* Measuring Subjective Sleep Quality: A Review. *Int. J. Environ. Res. Public.*

Health **18**, 1082 (2021).

13. Löwe, B., Kroenke, K., Herzog, W. & Gräfe, K. Measuring depression outcome with a brief self-report instrument: sensitivity to change of the Patient Health Questionnaire (PHQ-9). *J. Affect. Disord.* **81**, 61–66 (2004).
14. Levis, B. *et al.* Accuracy of the PHQ-2 Alone and in Combination With the PHQ-9 for Screening to Detect Major Depression: Systematic Review and Meta-analysis. *JAMA* **323**, 2290–2300 (2020).
15. Jaeger, J. Digit Symbol Substitution Test: The Case for Sensitivity Over Specificity in Neuropsychological Testing. *J. Clin. Psychopharmacol.* **38**, 513 (2018).
16. Alexander-Bloch, A. F. *et al.* On testing for spatial correspondence between maps of human brain structure and function. *NeuroImage* **178**, 540–551 (2018).

Reviewer #1 (Remarks to the Author):

The authors have made substantial changes to their manuscript, and I don't have any more comments.

Reviewer #2 (Remarks to the Author):

All my concerns have been addressed.